

# BingleSeq: a user-friendly R package for bulk and single-cell RNA-Seq data analysis

Daniel Dimitrov and Quan Gu

MRC-University of Glasgow Centre for Virus Research, University of Glasgow, Glasgow, UK

## ABSTRACT

**Background:** RNA sequencing is an indispensable research tool used in a broad range of transcriptome analysis studies. The most common application of RNA Sequencing is differential expression analysis and it is used to determine genetic loci with distinct expression across different conditions. An emerging field called single-cell RNA sequencing is used for transcriptome profiling at the individual cell level. The standard protocols for both of these approaches include the processing of sequencing libraries and result in the generation of count matrices. An obstacle to these analyses and the acquisition of meaningful results is that they require programing expertise. Although some effort has been directed toward the development of user-friendly RNA-Seq analysis analysis tools, few have the flexibility to explore both Bulk and single-cell RNA sequencing.

**Implementation:** BingleSeq was developed as an intuitive application that provides a user-friendly solution for the analysis of count matrices produced by both Bulk and Single-cell RNA-Seq experiments. This was achieved by building an interactive dashboard-like user interface which incorporates three state-of-the-art software packages for each type of the aforementioned analyses. Furthermore, BingleSeq includes additional features such as visualization techniques, extensive functional annotation analysis and rank-based consensus for differential gene analysis results. As a result, BingleSeq puts some of the best reviewed and most widely used packages and tools for RNA-Seq analyses at the fingertips of biologists with no programing experience.

**Availability:** BingleSeq is as an easy-to-install R package available on GitHub at https://github.com/dbdimitrov/BingleSeq/.

Corresponding author
Quan Gu, quan.gu@glasgow.ac.uk

## INTRODUCTION

About a decade ago, a transcriptome profiling approach, known as RNA Sequencing (RNA-Seq), was predicted to revolutionize transcriptome analyses (*Wang, Gerstein & Snyder, 2009*). Today, as a consequence of the continuous advancements and dropping costs of next-generation sequencing (NGS) technologies, differential expression (DE) analysis or Bulk RNA-Seq has established itself as a routine research tool (*Stark, Grzelak & Hadfield, 2019*).

Single-cell RNA Sequencing (scRNA-Seq) is an emerging field that enables the gene expression profiling at the individual cell level and it is believed to lead to the reconstruction of an entire human cell lineage tree (*Shapiro, Biezuner & Linnarsson, 2013*). scRNA-Seq's potential is highlighted by its broad range of applications which include the classification and discovery of cell subpopulations into putative transcriptomic profiles and novel cell types (*Jaitin et al., 2014*; *Macosko et al., 2015*; *Muraro et al., 2016*; *Campbell et al., 2017*; *Villani et al., 2017*), as well as the deconvolution of Bulk RNA-Seq results (*Schelker et al., 2017*; *Tirosh et al., 2016*).

Although there is a wide range of software tools available for both Bulk RNA-Seq and scRNA-Seq analyses, most require some proficiency in programing languages such as R. This creates a challenge for the analysis of RNA-seq data for a large portion of biologists lacking programing experience. Here we present an application, called BingleSeq, the primary goal of which is to enable the user-friendly analysis of count tables obtained by both Bulk RNA-Seq and scRNA-Seq protocols.

## MATERIALS AND METHODS

### Implementation

BingleSeq is based on shiny (*Chang et al., 2016*) and it is composed of a multi-tabbed UI, built as separate shiny modules with efficiency, code readability and reusability in mind. Each module (tab) corresponds to a key step in the typical Bulk and scRNA-Seq analysis pipelines (Fig. 1). Modules are generated only upon reaching a given step of the analyses which ensures efficiency and speed despite the complexity of the application. BingleSeq's UI components (e.g., plots, tables, tabs, etc.) make use of shiny's "reactivity" property and these components are automatically updated upon user input or any change related to a given component. The development of BingleSeq focused on providing a flexible and intuitive user experience. As such, BingleSeq implements three state-of-the art package options for each analysis, while it also enables users to tailor key analysis parameters according to their experiment.

User experience is complemented with customizable and interactive tables and plots as well as analysis-related tips and pop-up messages to guide the correct execution of the pipelines. Plot interactivity was achieved via conversion from ggplot to ggplotly (*Sievert, 2020*).

### Bulk RNA-seq steps and features

To begin the DE analysis of Bulk RNA-Seq data, a count table and a metadata table must be loaded in the appropriate formats (Fig. S1). Genes can then be filtered according to counts per million (CPM), Max, or Median thresholds. Furthermore, batch effect correction can be performed with Harman and ComBat packages (*Leek et al., 2012*; *Oytam et al., 2016*) (Fig. S2).

Subsequent to quality control, users can investigate differentially expressed genes (DEGs) using three state-of-the-art packages: DESeq2 (*Love, Huber & Anders, 2014*), edgeR (*Robinson, McCarthy & Smyth, 2010*), and limma (*Ritchie et al., 2015*).

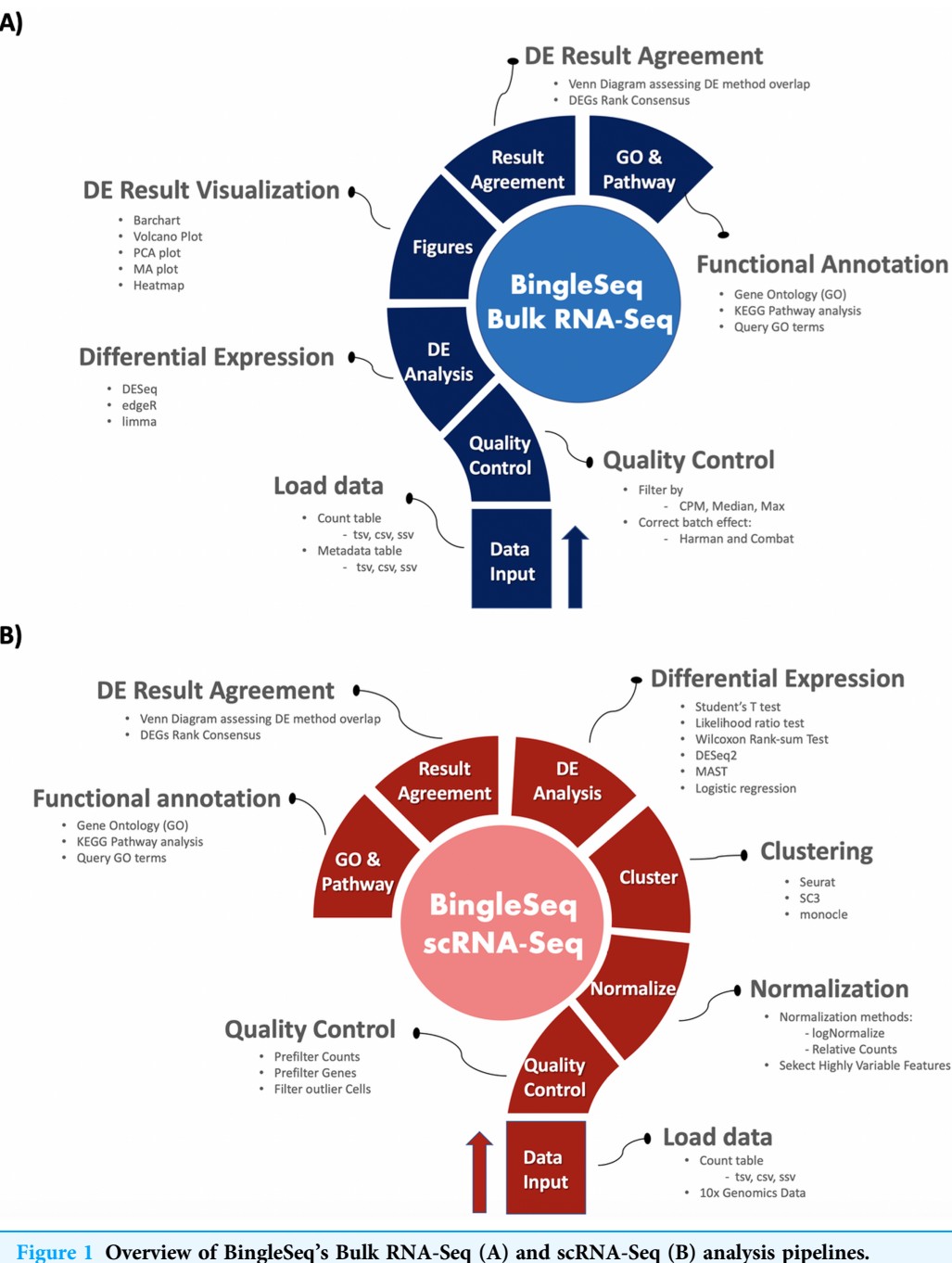

**Figure 1 Overview of BingleSeq's Bulk RNA-Seq (A) and scRNA-Seq (B) analysis pipelines.**

Upon obtaining DE results, users can visualize them using interactive plotting techniques (Figs. 2A–2E). These include: a PCA plot used to provide insights about the relationship between samples; Barchart plot supplemented with a summary table used to summarize the up- and downregulated DEGs; Volcano and MA plots that assess the relationship between fold change (FC) versus significance and average expression; and Heatmaps which are arguably the most versatile and informative type of visualization technique when looking at DE results.
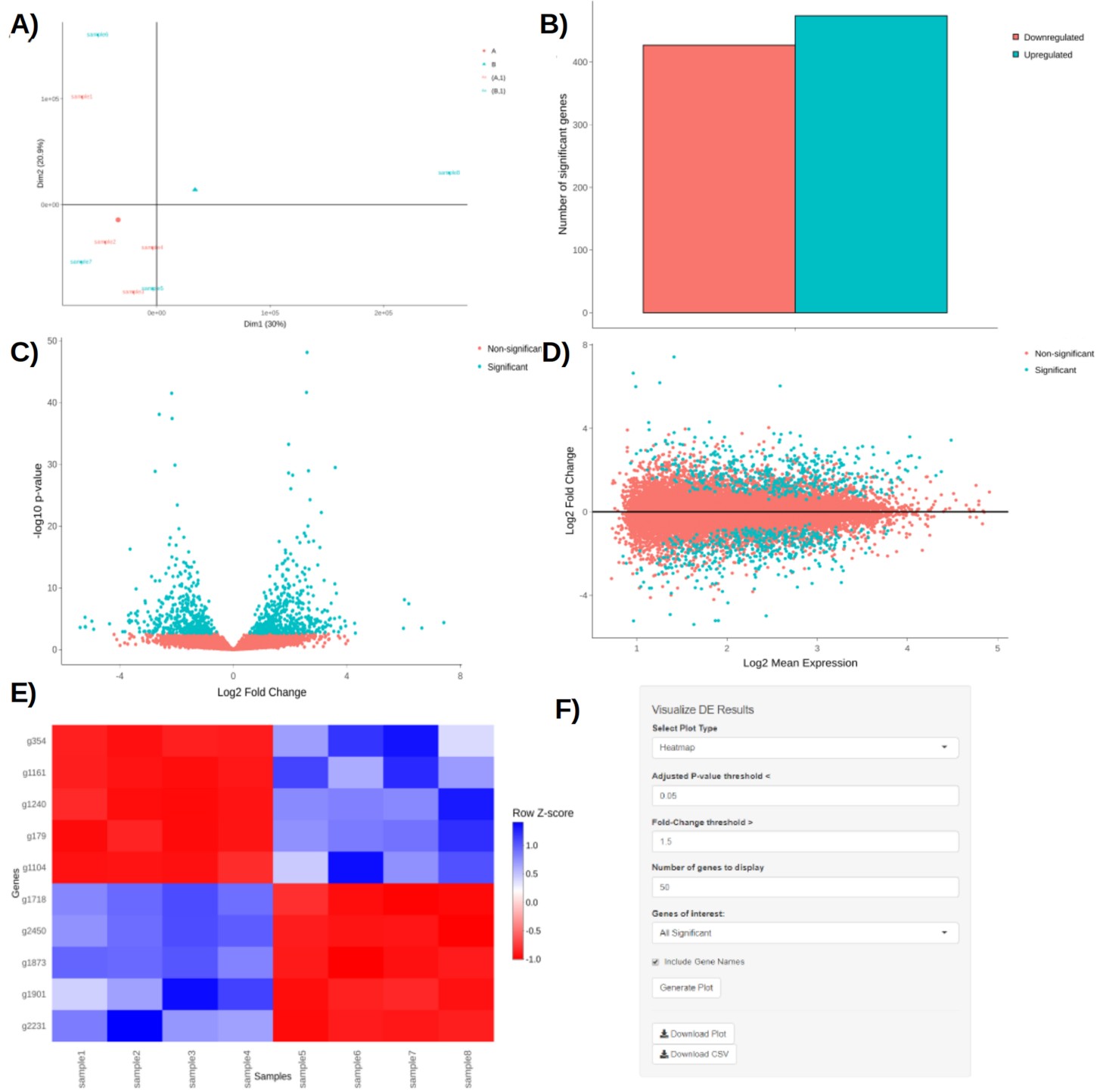

**Figure 2** PCA plot (A), barchart (B), volcano plot (C), MA plot (D), and a heatmap (E) with its corresponding interactive control panel (F) as generated by BingleSeq.

BingleSeq's visualization techniques were implemented with customization in mind and users can specify parameters such as *p*-value threshold and fold-change threshold, among others. Due to their versatility, heatmaps were designed as BingleSeq's most customizable plotting component (Fig. 2F).

To assess BingleSeq's Bulk RNA-Seq pipeline, we used a synthetic dataset generated with the compcodeR package (*Soneson, 2014*). We also tested it on a real data set looking at DEGs between HSV-1 infected control and interferon B treatment (*McFarlane et al., 2019*)—see Article S1.

## scRNA-seq pipeline steps and features

BingleSeq's scRNA-Seq part is based on Seurat's scRNA-Seq pipeline and visualizations (*Satija et al., 2015*). Nonetheless, clustering can also be performed with monocle (*Trapnell et al., 2014*) and SC3 (*Kiselev et al., 2017*) packages.

To begin scRNA-Seq analysis, data can be supplied in two formats—"Cell Ranger 10× Genomics" data and a count table in a predefined format (Fig. S3). Once the data is loaded, users can filter unwanted cells and features (Fig. S4). The next step is to normalize the data and BingleSeq provides two Seurat-supplied normalization methods—"LogNormalize" and "Relative counts". Simultaneously with normalization and scaling, the highly variable features within the dataset are identified and selected for clustering with Seurat as a way to minimize noise. The highly variable features are visualized using an interactive scatter plot (Fig. S4).

Following normalization, the "Clustering" tab is generated (Fig. 3) which provides a high degree of control over the different steps of the analysis (Fig. 3A). Additionally, general tips are provided, such as clustering advice provided for each package (Fig. 3B). Moreover, each of the visualization techniques in this tab are interactive to further facilitate the interpretation of results. First, the clustering tab requires the generation of pre-clustering prerequisites such as scaling the data and dimensionality reduction with Principal Component Analysis (PCA). Then an elbow plot is returned which is used to determine the dimensionality of the dataset (Fig. 3C)—this is essential for excluding noise when clustering with Seurat and monocle. PC heatmaps (Fig. 3D) are also available as a further tool for PC Selection. Once the data is filtered and transformed, users can proceed to unsupervised clustering with Seurat, SC3, and monocle. The primary way to visualize clustering results is via t-distributed stochastic neighbor embedding (tSNE) plots (Figs. 3E–3G)—a method designed for the purpose of visualizing high dimensional datasets (*Van der Maaten & Hinton, 2008*).

Following clustering, DE analysis can be conducted using Seurat's inbuilt testing methods to identify marker genes. The implemented Seurat DE methods include: Student's *T* test, Wilcoxon Rank Sum test, DESEq2 (*Love, Huber & Anders, 2014*), and MAST package (*Finak et al., 2015*). DE results and specific marker genes can then be visualized using Seurat's inbuilt plots. These plots include a cluster heatmap and visualizations for the exploration of specific genes via Violin, Feature, and Ridge plots (Fig. 4).

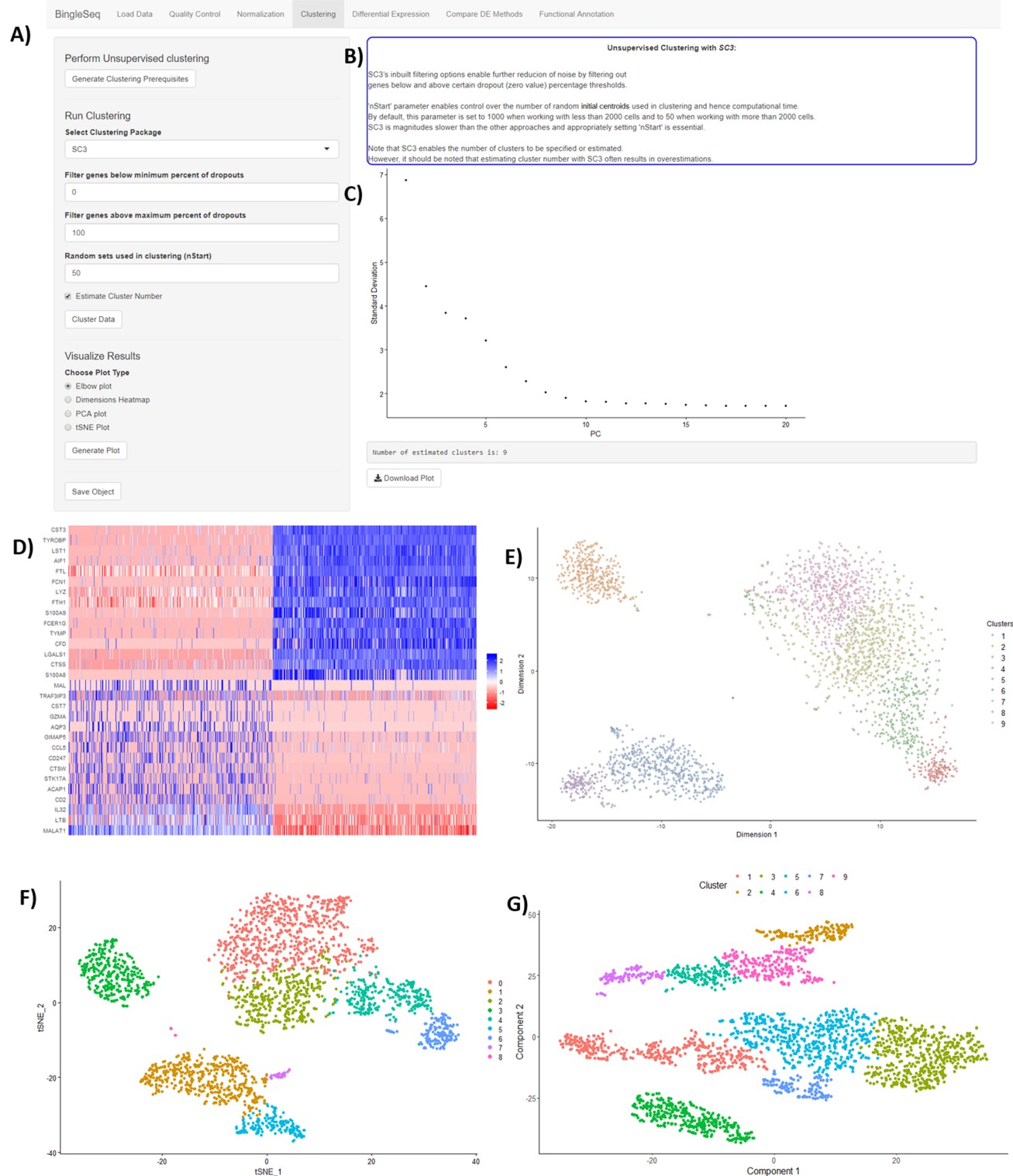

**Figure 3** An overview of BingleSeq's "Clustering" tab with (A) clustering customization options, (B) general tips and advice for the selected package, (C) PC elbow plot, (D) PC heatmap. tSNE plots from Seurat (E), SC3 (F), and monocle (G).

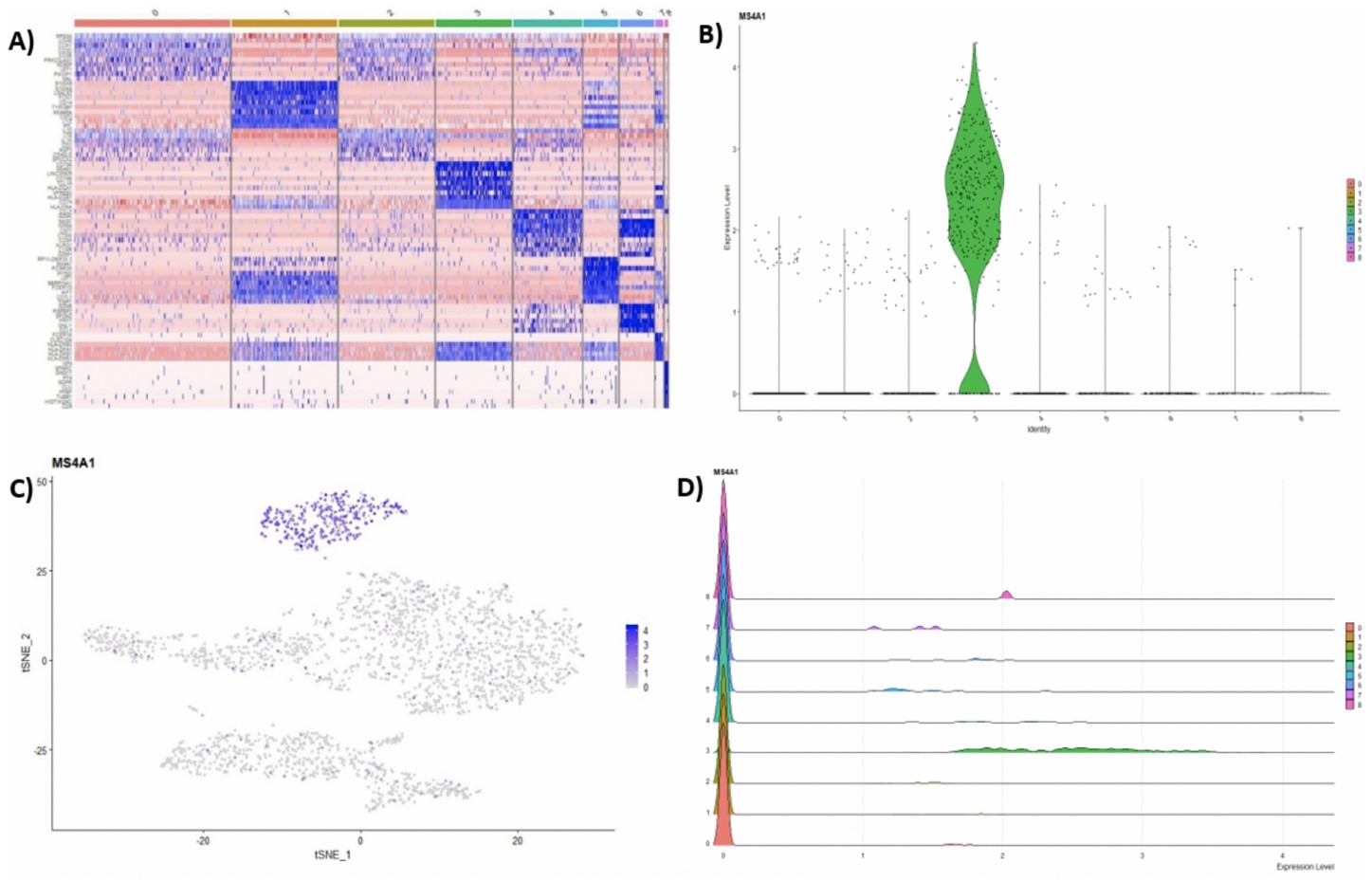

**Figure 4** (A) Heatmap showing the top 10 genes for each cluster in the 2,700 PBMCs dataset, while Violin (B), feature (C), and ridge (D) plots are shown for MS4A1 gene—a biomarker of B lymphocytes.

An evaluation of BingleSeq's scRNA-Seq pipeline was performed by reproducing and extending the results of Seurat's online tutorial (https://satijalab.org/seurat/v3.0/pbmc3k_tutorial.html)—see Article S2. The tutorial is based on a 10× Genomics dataset of 2,700 Peripheral Blood Mononuclear Cells (PBMCs) with ~69,000 reads per cell. The data set is available at https://support.10xgenomics.com/single-cell-gene-expression/datasets/1.1.0/pbmc3k.

## Functional annotation

For both Bulk and scRNA-Seq pipelines, BingleSeq enables the functional annotation of results using GOseq (*Young et al., 2010*) and transcription factor (TF) and pathway activity inference tools DoRothEA (*Garcia-Alonso et al., 2019*) and PROGENy (*Schubert et al., 2018*; *Holland, Szalai & Saez-Rodriguez, 2019*; *Holland et al., 2020*).

GOSeq is implemented in the "Gene Ontology" tab and it enables users to obtain results from KEGG pathway analysis and three types of GO categories, including "Cellular Component", "Molecular Function", and "Biological Function" (Fig. S5). The "Gene Ontology" tab can also be used to generate top 10 GO term histograms and to obtain

additional information about a given GO term using the "GO.db" package (*Carlson et al., 2019*) (Fig. S5). Note that BingleSeq supports human, mouse, drosophila, zebrafish, and *Escherichia coli* K12 strain genomes (*Carlson, 2019a*, *2019b*, *2019c*, *2019d*, *2019e*) for analysis with GOSeq.

In BingleSeq, the signed TF-target interactions from DoRothEA are coupled to the analytic Rank-based Enrichment Analysis (aREA) method from the viper package (*Alvarez et al., 2016*) and together they enable TF activity estimation (Figs. S6A and S6B). On the other hand, PROGENy uses a linear model to estimate the activity of 14 signaling pathways, built on downstream expression patterns from a compendium of perturbation experiments (Figs. S6C and S6D). DoRothEA and PROGENy are footprint-based tools and they infer activity from the expression of molecules considered to be downstream of a given TF or pathway, respectively. Both of these tools are implemented in the "Footprint Analysis" tab and are available for human and mouse data.

### DE package comparison and rank-based consensus

BingleSeq supplies an option to assess the agreement between the different DE analysis methods in the form of a Venn diagram (Fig. S7A). Moreover, BingleSeq provides a rank-based consensus approach to improve confidence in DE results (*Costa-Silva, Domingues & Lopes, 2017*; *Guo et al., 2014*; *Moulos & Hatzis, 2015*)—see Fig. S7B. In the context of the Bulk RNA-Seq pipeline the overlap in DEGs is assessed on the results obtained using DESeq2, edgeR, and limma packages. In the case of scRNA-Seq, this is done using three of Seurat's inbuilt DE methods—MAST, Wilcoxon Rank Sum Test, and Student's *T* test.

### Inbuilt bulk and single cell RNA-seq example datasets

BingleSeq also provides inbuilt Bulk RNA-Seq and scRNA-Seq test data. Bulk RNA-Seq data is represented by a 3-sample contrast between HSV-1 infected control and interferon B treatment (*McFarlane et al., 2019*). The data files are available from European Nucleotide Archive (ENA) under accession number PRJEB27501. The single-cell RNA-Seq example is a Cell Ranger 10× Genomics dataset looking at filtered data of 2,700 human peripheral blood mononuclear cells.

## RESULTS

Some effort has already been directed towards lowering the entry requirements to RNA-Seq analyses as there are some software tools which implement UI components. However, many of these applications are limited to only some key features or particular parts of RNA-Seq analysis (*DeTomaso & Yosef, 2016*; *Kiselev et al., 2017*).

From the available software packages providing comprehensive solutions for Bulk RNA-Seq, NetworkAnalyst (*Zhou et al., 2019*), DEapp (*Li & Andrade, 2017*), DEBrowser (*Kucukural et al., 2019*), and Omics Playground (*Akhmedov et al., 2020*) were thought to provide the most extensive RNA-Seq analysis options. As seen in Tables S1 and S2, the functionality implemented BingleSeq's Bulk RNA-Seq can be argued to put our package close to even the best available Bulk RNA-Seq solutions.

When looking at similar applications providing solutions to scRNA-Seq analysis, BingleSeq is most analogous to SeuratWizard (*Yousif et al., 2020*) as both are based on Seurat's pipeline. However, by implementing SC3 and monocle, BingleSeq provides solutions to some of Seurat's inherent limitations. For instance, SeuratWizard does not implement functionality to explicitly specify the number of clusters, nor a way to estimate the number of clusters, while BingleSeq provides two distinct approaches to achieve that. Another major functionality that sets our application apart is that it enables functional annotation analysis and a way to compare and provide a consensus for Seurat's inbuilt DE methods. Consequently, BingleSeq can be argued to match even the most comprehensive scRNA-Seq applications, such as ASAP (*Gardeux et al., 2017*) and singleCellTK (*Jenkins et al., 2020*)—see Tables S2 and S3.

In terms of providing a solution to both Bulk and single-cell RNA-Seq analyses, BingleSeq's features and comprehensiveness are only contested by those of Omics Playground. However, what sets BingleSeq apart is that it provides multiple clustering packages and algorithms.

## DISCUSSION

As solutions to Bulk RNA-Seq, BingleSeq implements DESeq2, edgeR, and limma. These packages are well-tested and regarded as being among the best performing ones (*Schurch et al., 2016*; *Seyednasrollah, Laiho & Elo, 2015*; *Soneson & Delorenzi, 2013*). Despite being accepted as being among the best DE analysis solutions, different studies often present contrasting conclusions. Hence, there is little consensus regarding which DE algorithm has the best performance. This stems from the fact that there is no optimal package under all circumstances and different variables are known to affect package performance, with sample size in particular (*Schurch et al., 2016*; *Soneson & Delorenzi, 2013*). As such, the method of choice depends on the dataset being analyzed.

BingleSeq's scRNA-Seq pipeline includes three unsupervised clustering solutions provided by monocle, Seurat, and SC3 packages. The latter two packages are regarded as having the best overall clustering performance (*Duò, Robinson & Soneson, 2018*; *Freytag et al., 2018*). However, similarly to packages used in the DE analysis of Bulk RNA-Seq data, there seems to be little consensus on which package provides the best-performing clustering approach. This is largely due to the inherent limitations of the different algorithms used in clustering, as a result no algorithm performs well in every circumstance (*Wiwie, Baumbach & Röttger, 2015*). *Kiselev, Andrews & Hemberg (2019)* suggest that Seurat may be inappropriate for small scRNA-Seq datasets, due to the inherent limitations of the Louvain algorithm. On the contrary, as a way to amend for the limitations of k-means clustering algorithm used in SC3, the authors implemented an extensive iterative-consensus approach, which makes SC3 magnitudes slower than Seurat and downgrades its scalability (*Duò, Robinson & Soneson, 2018*; *Kiselev, Andrews & Hemberg, 2019*). Another difference between these two packages is that Seurat does not include functionality to estimate or explicitly specify cluster number, while SC3 does.

Thus, by implementing multiple DE and clustering packages BingleSeq enables users to explore and pick the method that is most suitable for their experiment.

## CONCLUSIONS

BingleSeq is a comprehensive and intuitive solution that enables users to choose from multiple state-of-the-art DE analysis and unsupervised clustering packages according to their preferences or the dataset in question. In terms of Bulk RNA-Seq analyses, BingleSeq implements functionality that puts it close to, what are to our understanding, the best available similar applications. In terms of scRNA-Seq, BingleSeq could be argued to be among the most exhaustive applications, such as ASAP (*Gardeux et al., 2017*) and singleCellTK (*Jenkins et al., 2020*). Thus, the implementation of multiple state-of-the-art packages for both types of analysis, alongside functionality that bolsters the confidence and subsequent interpretation of DE results, makes BingleSeq a solid choice for the analysis of both Bulk and scRNA-Seq.

Future work will focus on including more functional annotation options and on extending user control over the packages implemented in BingleSeq. Moreover, the implementation of both Bulk RNA-Seq and scRNA-Seq pipelines puts BingleSeq in a particularly good position to implement user-friendly deconvolution of Bulk RNA-Seq results using scRNA-Seq data. Hence, an excellent and practical conclusion to the development of BingleSeq would be to include state-of-the-art deconvolution methods such as Cell Population Mapping (CPM) (*Frishberg et al., 2019*) or MUlti-Subject SIngle Cell deconvolution (MuSiC) (*Wang et al., 2019*).

BingleSeq is as an easy-to-install R package available on GitHub at https://github.com/dbdimitrov/BingleSeq/. The application's GitHub page provides an easy guide on how to install the application as well as examples of its general applicability and an extensive description of typical workflows when working with Bulk RNA-Seq and scRNA-Seq data.

## ACKNOWLEDGEMENTS

We also extend our gratitude to Sejal Modha, Joseph Hughes, Richard Orton, Srikeerthana Kuchi, and Martín Garrido-Rodríguez for reviewing the manuscript prior to submission.

### Funding

This work was financially supported by bioinformatics developments as part of MRC (MC_UU_12014/12). The funders had no role in study design, data collection and analysis, decision to publish, or preparation of the manuscript.

### Grant Disclosures

The following grant information was disclosed by the authors:
MRC: MC_UU_12014/12.

### Competing Interests

The authors declare that they have no competing interests.

## Author Contributions

- Daniel Dimitrov performed the experiments, analyzed the data, prepared figures and/or tables, authored or reviewed drafts of the paper, and approved the final draft.
- Quan Gu conceived and designed the experiments, analyzed the data, authored or reviewed drafts of the paper, and approved the final draft.

## Data Availability

Bulk RNA-Seq data is represented by a 3-sample contrast between HSV-1 infected control and interferon B treatment (*McFarlane et al., 2019*).

The data files are available from European Nucleotide Archive: PRJEB27501.

The single-cell RNA-Seq example is a Cell Ranger 10× Genomics public dataset looking at filtered data of 2,700 Peripheral Blood Mononuclear Cells (PBMCs) with ~69,900 reads per cell (https://support.10xgenomics.com/single-cell-gene-expression/datasets/1.1.0/pbmc3k) and requires user registration to access. It is also available at GitHub: https://github.com/dbdimitrov/BingleSeq/tree/master/inst/extdata.

The R package source code is also available at GitHub: https://github.com/dbdimitrov/BingleSeq.

## Supplemental Information

Supplemental information for this article can be found online at http://dx.doi.org/10.7717/peerj.10469#supplemental-information.

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
