# Peer review of "BingleSeq: a user-friendly R package for bulk and single-cell RNA-Seq data analysis"

_PeerJ, doi:10.7717/peerj.10469_

## Round 0.1 · original submission · Minor Revisions

I have a few comments in addition to Reviewer 1:

- The font sizes in the visualizations created by BingleSeq seem consistently too small. As an example, consider your readme here:
https://github.com/dbdimitrov/BingleSeq#4visualization
The figure labels should be at least as large as the font used for the text of the readme. Currently they are way smaller.

- I encourage you to consider submitting your package to CRAN or Bioconductor. Experience shows that packages that are available through one of these channels see much more use than packages that need to be installed from source.

- In your readme, I'm not sure why you provide a long and complicated installation procedure when install_github() is sufficient.

Reviewer 1 ·

Basic reporting

Basic
1. The Materials & Methods section contains far too much commentary that is more appropriate for the Results & Discussion section. Examples include lines 85 to 92 and all of the Related Applications subsection.
2. The phrasing is at times awkward and grammatically confusing. Examples include lines 16-17, 113-116, and 150-152. Careful proofreading and editing for clarity would improve the manuscript.

Experimental design

no comment

Validity of the findings

no comment

Additional comments

General
1. The application itself is well designed and offers a great tool for RNAseq analysis.
2. Some minor application improvements could be made, including:
- Adding the ability to select other genomes outside of the mouse and human ones included for gene enrichment analysis
- Using plotly (via ggplotly) or another package to provide interactive plots in some sections to facilitate exploration of results.
- Adding a filterable gene data column (right now gene ids are provided as row ids)
- More information on longer loading screens, e.g. showing cluster progressing during differential expression analysis of scRNAseq data.

---

## Round 0.2 · accepted · Accept

Thank you for your careful revision.